# Synergistic Adverse Effects of Azithromycin and Hydroxychloroquine on Human Cardiomyocytes at a Clinically Relevant Treatment Duration

**DOI:** 10.3390/ph15020220

**Published:** 2022-02-12

**Authors:** Wener Li, Xiaojing Luo, Mareike S. Poetsch, Reinhard Oertel, Kapil Nichani, Martin Schneider, Anna Strano, Marcel Hasse, Robert-Patrick Steiner, Lukas Cyganek, Karina Hettwer, Steffen Uhlig, Kirsten Simon, Kaomei Guan, Mario Schubert

**Affiliations:** 1Institute of Pharmacology and Toxicology, Technische Universität Dresden, 01307 Dresden, Germany; wener.li@tu-dresden.de (W.L.); xiaojing.luo@tu-dresden.de (X.L.); mareike.poetsch@tu-dresden.de (M.S.P.); anna.strano@tu-dresden.de (A.S.); marcel.hasse@tu-dresden.de (M.H.); robert-patrick.steiner@tu-dresden.de (R.-P.S.); 2Institute of Clinical Pharmacology, Technische Universität Dresden, 01307 Dresden, Germany; reinhard.oertel@tu-dresden.de; 3QuoData–Quality & Statistics GmbH, 01309 Dresden, Germany; kapil.nichani@quodata.de (K.N.); martin.schneider@quodata.de (M.S.); hettwer@quodata.de (K.H.); uhlig@quodata.de (S.U.); simon@quodata.de (K.S.); 4Clinic for Cardiology and Pneumology, University Medical Center Göttingen, 37075 Göttingen, Germany; lukas.cyganek@med.uni-goettingen.de; 5German Center for Cardiovascular Research (DZHK), Partner Site Göttingen, 37075 Göttingen, Germany

**Keywords:** hydroxychloroquine, azithromycin, field potential duration, conduction velocity, human induced pluripotent stem cells, cardiomyocytes, drug testing, drug interaction

## Abstract

Adverse effects of drug combinations and their underlying mechanisms are highly relevant for safety evaluation, but often not fully studied. Hydroxychloroquine (HCQ) and azithromycin (AZM) were used as a combination therapy in the treatment of COVID-19 patients at the beginning of the pandemic, leading to higher complication rates in comparison to respective monotherapies. Here, we used human-induced pluripotent stem cell-derived cardiomyocytes (iPSC-CMs) to systematically investigate the effects of HCQ, AZM, and their combination on the structure and functionality of cardiomyocytes, and to better understand the underlying mechanisms. Our results demonstrate synergistic adverse effects of AZM and HCQ on electrophysiological and contractile function of iPSC-CMs. HCQ-induced prolongation of field potential duration (FPDc) was gradually increased during 7-day treatment period and was strongly enhanced by combination with AZM, although AZM alone slightly shortened FPDc in iPSC-CMs. Combined treatment with AZM and HCQ leads to higher cardiotoxicity, more severe structural disarrangement, more pronounced contractile dysfunctions, and more elevated conduction velocity, compared to respective monotreatments. Mechanistic insights underlying the synergistic effects of AZM and HCQ on iPSC-CM functionality are provided based on increased cellular accumulation of HCQ and AZM as well as increased Cx43- and Nav1.5-protein levels.

## 1. Introduction

Cardiotoxicity represents one of the top reasons for drug withdrawal from clinical trials and the market, with increasing attrition rates over the past decades [1]. One of the major cardiotoxic effects is fatal cardiac arrhythmia due to direct drug interactions with cardiac electrophysiology. Therefore, preclinical testing for cardiac adverse effects has been focused primarily on the investigation of ion channel functions and their associated abnormalities in action potentials and conduction velocity. However, many studies reveal that mechanisms of cardiotoxicity are more complex, go beyond effects on cardiac electrophysiology [1,2], and involve dysregulation of a variety of cellular processes such as calcium cycling, cellular trafficking, mitochondrial function, and apoptosis [1]. Importantly, consequences of impaired trafficking or apoptosis may establish during prolonged exposure to a drug, especially if drug accumulation occurs even though the compound is applied in a rather low, clinically relevant concentration.

At the beginning of the worldwide pandemic caused by the severe acute respiratory syndrome coronavirus 2 (SARS-CoV-2), attempts using hydroxychloroquine (HCQ) in combination with azithromycin (AZM) reported first positive results for the treatment of SARS-CoV-2 infected (COVID-19) patients, demonstrating reinforced viral load reduction/disappearance in a small number of COVID-19 patients [3,4]. However, the follow-up clinical trials could not confirm the efficacy of treatment with HCQ or HCQ in combination with AZM [5,6,7]. QT interval prolongation, as reported by various groups [8,9,10,11], is considered as an adverse effect of the combined treatment with HCQ and AZM in COVID-19 patients.

Both HCQ and AZM are known to accumulate in cells, impair important cellular processes such as autophagy [12,13], and to affect cardiac electrophysiology. Conduction disorders were reported to occur in 85% of patients after chronic treatment with HCQ (or chloroquine) and represented one of the main side effects of HCQ [14,15]. Mechanistic insights from animal models revealed that acute application of HCQ reduces the heart rate by modulating the funny current *I_f_* [16,17]. AZM, a broad-spectrum macrolide antibiotic, was considered a good safety profile until the report of a small absolute increase in cardiovascular deaths during 5 days of AZM therapy [18]. In addition, several cases of AZM-induced QT-interval prolongation were reported in the clinic [17,19]. So far, cardiotoxicity of HCQ and AZM was mainly studied in acute or short-term experimental settings [16,17,20,21,22]. There is limited knowledge about their interactions and effects on the function of human cardiomyocytes when applied in combination in a clinically relevant concentration range and duration, which may contribute to the clinically observed complications. As the mechanisms underlying HCQ- and AZM-related cardiac synergistic effects are not fully understood, the benefit-risk balance of treating COVID-19 patients with these compounds remains a dilemma for physicians.

The aim of the study was to investigate the effects of HCQ, AZM, and their combination, to identify potential synergistic drug interactions and to better understand their arrhythmia-inducing mechanisms in an in vitro human cardiomyocyte model system. Clinical practice in the treatment of COVID-19 patients and previously reported plasma concentrations of HCQ and AZM served as a rationale for the design of our study. Drug concentrations ranging from 1 to 10 µM were defined based on the antiviral potency of HCQ (EC_50_: 4.2 µM) and AZM (EC_50_: 2.1 µM) [23] and the reported dosages used for COVID-19 patients (600–800 mg/day for HCQ and 250–500 mg/day for AZM). The therapeutic blood levels of HCQ for systemic lupus erythematosus were 1.5 µM to 6 µM in patients receiving a dose of 200 or 400 mg/day [24,25]. Although AZM plasma level was rather low, ~0.3 µM in patients receiving a dose of 250 mg daily [22,26], AZM is known to accumulate rapidly in cells [22,27,28,29,30,31]. The use of induced pluripotent stem cell-derived cardiomyocytes (iPSC-CMs) in this study offered a robust platform to investigate the consequences of HCQ and AZM treatment on the viability, the contractile structure, contractility, and electrophysiology of human cardiomyocytes. To investigate the effects of the drugs on human cardiomyocytes reflecting different genetic backgrounds, iPSC-CMs from four different donors (1–2 iPSC lines each) without known cardiovascular disease were treated with HCQ, AZM, or their combination for 7 days (Appendix A). Afterward, they were cultured for another 7 days without the drugs (washout period). The treatment duration of 7 days was chosen based on the time course of HCQ and AZM treatment in COVID-19 patients, which varied between 5 and 10 days [3,6,9,32]. The duration of the washout phase was determined based on clinical case reports describing the reversibility of HCQ-induced QT prolongation 4–7 days after drug withdrawal [33,34].

## 2. Results

### 2.1. Effects of HCQ and AZM on Cell Morphology and Viability

First, the effects of HCQ and AZM on the morphology of iPSC-CMs were investigated. Treatments with AZM and HCQ, in particular at higher concentrations, caused the formation of vacuole-like structures within the cells (Figure 1A and Appendix A), which persisted until 7 days after drug washout (Figure 1D and Appendix A). Overview images (Appendix A) and cell nucleus counting (Appendix A) showed reduction in total cell number after 7 days of the combined treatment with 10 µM HCQ and 10 µM AZM. Importantly, treatment with vehicle had no influence on viability or cell number (Appendix A). Cells treated with 10 µM HCQ alone or in combination with 1 µM AZM showed a progressive cell death (Figure 1B). The MTT assay revealed that 7-day combination treatment with 10 µM HCQ and 10 µM AZM led to less than 50% of cells at a viable and metabolically active state (Figure 1B), whereas HCQ (1 µM or 3 µM) alone or in combination with AZM (1 µM or 10 µM), respectively, did not significantly affect metabolic activity of iPSC-CMs (Figure 1B). Significantly higher rates of cell death were also observed based on increased lactate dehydrogenase (LDH) activity in the cell supernatant of the groups treated with 10 µM HCQ alone and in combination with AZM (1 µM or 10 µM) (Figure 1C). In contrast to HCQ, 10 µM AZM alone showed no effect on cell viability using both MTT and LDH activity assays (Figure 1B,C).

After 7-day drug washout, AZM-induced accumulation of vacuole-like structures was reduced, consistent with a previously published study [13]. However, the cytotoxic effect of 10 µM HCQ became more evident. The MTT assay revealed a further decrease in viability of cells treated with 10 µM HCQ alone or in combination with 1 µM AZM, which was consistent with reduced cell confluence (Figure 1D,E). Notably, due to significantly reduced cell numbers in groups treated with 10 µM HCQ alone or in combination with AZM as well as the daily medium change, LDH activity in the supernatant is not representative in these samples after the drug washout (Figure 1F). Treatment of cells with lower drug concentrations (1 or 3 µM HCQ, 1 or 10 µM AZM) did not affect cell viability (Figure 1E,F). These results demonstrate the toxicity of HCQ at high concentrations, which is further increased in the presence of AZM.

### 2.2. HCQ and AZM Affect the Structural Organization of iPSC-CMs

To investigate the effects of HCQ and AZM on iPSC-CM area, sarcomere organization, and sarcomere length, we performed immunofluorescence staining to detect α-actinin. To evaluate the effect of AZM and HCQ on cell area, iPSC-CMs were seeded at low density to monitor single cells. Single cells were less resistant to drug treatment compared to cells in monolayer by showing severe morphological changes and cell death, in particular, under treatment with 10 µM HCQ and 10 µM AZM either alone or in combination (Figure 2A). Therefore, structural analyses of iPSC-CMs were only performed for treatments with lower drug concentrations, for which cell detachment was less evident.

The 7-day treatment with 1 µM AZM alone resulted in a slight increase in cell area (Figure 2B), but this did not persist, and rather resulted in a slight decrease after drug washout (Figure 2C). After 7-day treatment with 3 µM HCQ alone, iPSC-CMs showed a reduction in cell area, which was not obvious in the groups treated with 1 µM HCQ alone or with HCQ (1 and 3 µM) in combination with 1 µM AZM (Figure 2A,B). However, after the drug washout, iPSC-CMs treated with 1 and 3 µM HCQ alone or in combination with 1 µM AZM showed smaller cell areas compared to control cells, indicating persistent cellular shrinking (Figure 2C).

To quantify the effect of HCQ and AZM on sarcomeric organization in iPSC-CMs, the proportion of cells with structurally organized and disorganized sarcomeres were manually determined based on the images of iPSC-CMs stained for α-actinin. Cells with evenly distributed intact sarcomeres across the cell body (occupying > 80% of the cell area) were classified as structurally organized (Figure 2D, left), while cells with intact sarcomeres distributed exclusively in the center or periphery or cells lacking clearly organized ladder-like sarcomeres were classified as structurally disorganized (Figure 2D, right). Under basal conditions, 61 ± 6% of iPSC-CMs were classified as structurally organized (Figure 2E). The relatively high portion of cells with disorganized sarcomeres at basal condition might result from the immaturity of iPSC-CMs undergoing sarcomere assembly. Treatment with 1 µM AZM and 1 µM HCQ alone revealed no effect on the sarcomere organization of iPSC-CMs. An increase in the percentage of structurally disorganized cells was found in cells treated with 3 µM HCQ alone (*p* = 0.055) or in combination with 1 µM AZM (*p* < 0.0001, Figure 2E).

As another important aspect of iPSC-CM structure, the sarcomere length was measured in the population of structurally organized cells (Figure 2D, left). The sarcomere length of iPSC-CMs at basal condition was determined as 2.04 ± 0.05 µm, which is comparable to a sarcomere length of ~2.2 µm observed in mature cardiomyocytes [35]. After a 7-day treatment with 1 µM AZM, 3 µM HCQ, or the combination of HCQ (1 and 3 µM) and 1 µM AZM, iPSC-CMs showed a significant reduction in sarcomere length, which was not obvious in the group treated with 1 µM HCQ alone (Figure 2F). The strongest reduction in sarcomere length was observed in the group treated with 1 µM AZM combined with 3 µM HCQ. After the subsequent washout period for 7 days, sarcomere length remained strongly reduced in groups treated with 3 µM HCQ alone and HCQ in combination with AZM and slightly reduced in iPSC-CMs treated with 1 µM AZM or 1 µM HCQ alone (Figure 2G).

Taken together, these results highlight the negative effect of HCQ and AZM treatments on the structural characteristics of iPSC-CMs and the persistence of their adverse effects even after drug washout for 7 days.

### 2.3. HCQ and AZM Alter the Contractility of iPSC-CMs

The effects of HCQ and AZM on the beating property of iPSC-CMs were investigated using video-based motion vector analysis (Appendix A). This method allows the quantification of specific parameters of contraction and relaxation [36,37]. As a quality control, all beating parameters remained unchanged for cells cultured with 0.1% DMSO (vehicle) during the 7-day treatment (Appendix A). A progressive alteration of cardiomyocyte beating properties was observed in the presence of AZM and HCQ during the 7-day treatment period (Figure 3 and Appendix A). While AZM (1 µM) and HCQ (1 µM and 3 µM) alone had no or less effect on the beating parameters of iPSC-CMs during the 7-day treatment (Figure 3B–E), strong changes were observed in iPSC-CMs treated with AZM (10 µM) and HCQ (10 µM) alone or with AZM (1 and 10 µM) in combination with HCQ (1, 3, and 10 µM) (Figure 3B–E and Appendix A). Notably, treatment with 10 µM HCQ alone or combined with 1 µM or 10 µM AZM led to stopping of beating or strongly distorted motion in some cultures of iPSC-CMs, which could not be included in the analysis (Figure 3A,B). With respect to beating rate (Figure 3B and Appendix A), iPSC-CMs treated with 10 µM AZM alone showed an increase in beating rate at day 1 but a decrease at day 7 (Figure 3B and Appendix A), while 10 µM HCQ alone progressively increased the beating rate of iPSC-CMs from day 1 onwards (Figure 3B and Appendix A). A combination of AZM (1 or 10 µM) with 10 µM HCQ led to even higher beating rates than 10 µM HCQ alone (Figure 3B). Moreover, an increased beating rate was observed in the group treated with 3 µM HCQ in combination with 10 µM AZM, which was absent in the cells treated with 3 µM HCQ alone (Figure 3B), indicating the synergistic effects of both drugs. In terms of contraction time and relaxation time, 10 µM AZM alone showed a progressive reduction, similar to the group treated with 10 µM HCQ alone during the 7-day treatment (Figure 3C,D and Appendix A). The combination of HCQ and AZM enhanced the decrease of contraction and relaxation time in a concentration- and time-dependent manner (Figure 3C,D).

Overall, these data demonstrate that AZM and HCQ directly affect the beating rate, as well as the contraction and relaxation behavior of iPSC-CMs in a concentration- and time-dependent manner, while the combination of AZM and HCQ enhances the effects of HCQ on increasing the beating rate as well as on decreasing contraction time and relaxation time of iPSC-CMs.

### 2.4. HCQ and AZM Lead to the Prolongation of Field Potential Duration in iPSC-CMs

To assess the effect of HCQ and AZM on the heart rhythm, the field potential (FP) analysis in iPSC-CMs were performed using the multi-electrode array (MEA) technique. As shown in Figure 4A,B, the corrected FP duration (FPDc) in the control group remained stable while 1 µM AZM showed no effect on the FPDc during the 14-day recording (7-day drug treatment and subsequent 7-day washout). However, 10 µM AZM slightly shortened the FPDc of iPSC-CMs, and drug washout could not restore it to the basal level (Figure 4A,B). Treatment with HCQ at low concentrations (1 µM and 3 µM) had no effect on FPDc, however, iPSC-CMs treated with 10 µM HCQ showed a prolonged FPDc from day 2, which kept rising until day 7 (Figure 4A,C). The prolongation of FPDc induced by 10 µM HCQ was reversible, as drug washout gradually eliminated this effect.

When 1 µM AZM was combined with HCQ (1, 3 or 10 µM), similar effects as HCQ alone were observed, showing the prolongation of FPDc only with 10 µM HCQ, but to a lesser extent (Figure 4D). Notably, the combination of 10 µM AZM with 3 µM HCQ significantly and reversibly prolonged the FPDc of iPSC-CMs, which was not observed in cells treated with the combination of 10 µM AZM and 1 µM HCQ (Figure 4). When we combined 10 µM AZM with 10 µM HCQ, the prolonged FPDc in iPSC-CMs was evident from day 3 to day 8 (Figure 4E). However, we observed that 55% of iPSC-CMs failed to reveal FP and showed cell death on day 8 (the first day of washout), and 82% of cultures stopped beating at the end of the experiment (Figure 4A,E and Appendix A). In terms of beating frequency, 10 µM AZM caused a significant increase in spontaneous beating frequency on day 1 but a lower beating frequency from day 4 onwards (Appendix A) whereas 10 µM HCQ led to a significant increase from day 1 onwards (Appendix A), which is in line with the results observed in the contractility experiments. Interestingly, most drug-treated iPSC-CMs had a slower beating rate than the control group during the washout period (Appendix A).

### 2.5. HCQ and AZM Independently and Synergistically Augment the Conduction Velocity of iPSC-CMs

Since conduction disorders were the most frequent side effect that appeared in COVID-19 patients who were administrated with HCQ and AZM [38], we examined the impact of the two drugs on cardiac conduction velocity (CV) in iPSC-CM model. As shown in Figure 5, CV of iPSC-CMs in the control group remained stable during the two-week experiment. While cells treated with 1 µM AZM showed a similar conduction trajectory and CV as in the control group, 10 µM AZM led to changes in trajectory and significantly augmented CV in iPSC-CMs, starting on day 3 after drug treatment, but reversing on day 3 after drug washout (Figure 5A,B). Similar to AZM, HCQ also resulted in changes in conduction trajectory and increases in the CV of iPSC-CMs in a concentration-dependent pattern (Figure 5A,C). The addition of 1 µM AZM enhanced the effects caused by HCQ alone (Figure 5D). Furthermore, when 10 µM AZM was applied in addition to HCQ (1, 3, and 10 µM), iPSC-CMs from all three groups showed significantly faster transmission of electrical signals (Figure 5A,E).

### 2.6. HCQ and AZM Synergistically Enhance the Expression of Cx43 and Alter the Steady-State Kinetics of I_Na_ in iPSC-CMs

To gain insights into the molecular mechanism of HCQ/AZM-induced CV augmentation, we analyzed expression of Nav1.5 and Cx43, which are crucial to maintaining electrical signal propagation between CMs [39]. Compared to the control group, the expression of Nav1.5 was slightly, but not significantly, higher in iPSC-CMs treated with 10 µM HCQ for 7 days (*p* > 0.05, Figure 6A,B). Treatment with 10 µM AZM did not change Nav1.5 protein levels (*p* > 0.05, Figure 6A,B). Importantly, when we applied 10 µM HCQ combined with 10 µM AZM to iPSC-CMs, we observed a two-fold increase in Nav1.5 protein expression (*p* = 0.08, Figure 6A,B). In terms of Cx43, 7-day treatment with 10 µM HCQ significantly increased the protein expression by three-fold (*p* < 0.01, Figure 6A,C). While treatment with 10 µM AZM alone only slightly increased the Cx43 expression (*p* > 0.05), the combination of 10 µM HCQ and 10 µM AZM synergistically quadrupled the expression of Cx43 compared to the control group (Figure 6A,C). Similar results were observed using immunofluorescence staining, revealing a higher expression as well as a strong intracellular accumulation of Cx43 in iPSC-CMs treated with 10 µM HCQ, 10 µM AZM, and their combination (Figure 6D).

To further investigate the impact of HCQ and AZM on the function of cardiac sodium channel, we recorded *I_Na_* in cells treated with 10 µM HCQ and/or 10 µM AZM for 7 days using an automated patch-clamp technique (Appendix A). Compared to the control group, iPSC-CMs treated with 10 µM AZM alone showed increased membrane capacitances (an indicator for cell size), while cells treated with 10 µM HCQ in combination with 10 µM AZM showed lower membrane capacitances (Appendix A). We could not observe differences regarding the current density of *I_Na_* in the four groups, except that the reversed current at +70 mV was smaller in cells treated with combined HCQ and AZM (Appendix A). However, both drugs markedly modified the gating properties of cardiac sodium channel. Compared to the control group, the steady-state activation curves were leftwards shifted in the groups treated with HCQ alone or in combination with AZM, but not in iPSC-CMs treated with AZM alone (Appendix A). Moreover, the steady-state inactivation curves of all the three groups treated with the drugs showed a rightwards shift (Appendix A).

### 2.7. HCQ and AZM Accumulate in iPSC-CMs

As HCQ and AZM have been reported to accumulate in lysosomes and endosomes, we analyzed the levels of HCQ and AZM in lysates of iPSC-CMs after a 7-day drug treatment using mass spectrometry. Due to reduced cell viability in combination treatments with higher concentrations of HCQ and AZM, we did not include these groups in the analysis. Our data reveal that cellular levels of HCQ (Figure 7A) and AZM (Figure 7B) increased in a concentration-dependent manner after the 7-day treatment. Interestingly, levels of HCQ in iPSC-CMs treated with 1 µM HCQ combined with 10 µM AZM were much higher than those in cells treated with 1 µM HCQ alone (Figure 7A) and accumulation of AZM was increased by co-treatment with 1 µM (*p* = 0.06) or 3 µM (*p* = 0.12) HCQ (Figure 7B). These data indicate that the combined treatments with HCQ and AZM facilitate cellular accumulation of HCQ and AZM and provide further evidence for the synergistic effects of AZM and HCQ through increased cellular accumulation.

## 3. Discussion

The HCQ/AZM combination therapy was associated with increased cardiac complication rates in comparison to monotherapy with HCQ or AZM [8]. Based on the known cardiac adverse effects of HCQ and AZM, and their accumulation properties, here we investigate their potential synergistic effects on iPSC-CM structure and function for a period of 7 days, similar to clinical treatment durations of 5–10 days [3,6,9,32,40].

In contrast to acute functional changes caused by inhibition of ion channels, cardiotoxic effects and reduced cell viability induced through mitochondrial dysfunction or impaired autophagy may establish during longer time scales. We show that both AZM and HCQ negatively affect the viability, morphology, and sarcomeric structure as well as the functionality of iPSC-CMs at clinically relevant concentrations and treatment duration. The progressively observed changes in contractile and electrophysiological parameters during 7 days of drug treatment, revealed by MEA-measurements and video-based motion analysis, provide first evidence for the functional consequences of AZM and HCQ under long-term treatment, whereas insights from previous studies are limited to acute or short-term treatment [20,21]. Interestingly, our data reveal synergistic effects of AZM and HCQ. The combination with AZM strongly increased HCQ-induced reduction of cell viability as well as changes in contractile and electrophysiological function. Moreover, we demonstrate that HCQ and AZM increased Cx43 and Nav1.5 protein levels in a synergistic manner, which may underlie the severe electrophysiological dysfunction. Mechanistic insights on the synergistic effect of HCQ and AZM are provided by the increased accumulation of the drugs in iPSC-CMs when applied in combination.

### 3.1. HCQ and AZM Differentially Affect iPSC-CM Viability and Functionality

In this study, treatments with AZM and HCQ alone revealed that both drugs at higher concentrations negatively impact cell viability, morphology, sarcomeric structure, the contractility, and electrophysiological function of iPSC-CMs. At an equimolar concentration of 10 µM, however, a significantly higher cardiotoxic activity of HCQ than that of AZM was observed, as shown by lower MTT reduction to formazan, lower cell density, and higher LDH activity after the 7-day treatment (Figure 1 and Appendix A). Even after 7 days of drug washout, a progressive cardiotoxic effect of HCQ was detected not only by the MTT assay, morphological analysis, but also by the increasing the number of iPSC-CM cultures that stopped spontaneous beating (Appendix A). Besides the reduced cell viability, treatment with 10 µM HCQ resulted in a progressive increase in FPDc, CV, and beating frequency in iPSC-CMs during the 7-day treatment. The increased FPDc was also reported in the guinea pig heart upon acute treatment with 10 µM HCQ alone ex vivo [20]. In our study, we observed a slight reduction of FPDc in iPSC-CMs after the 7-day treatment with 10 µM AZM alone, while CV was increased to a similar extend as in cells treated with 10 µM HCQ. Interestingly, AZM led to an initial increase in the beating frequency on day 1, but a decrease to control levels on day 3 and a further decrease until day 7. The AZM-induced increase in the beating rate at day 1 is in line with the previous study showing that treatment of HL-1 CMs with 100 µM AZM for 24 h dramatically increased the spontaneous beating frequency [22]. Although several studies reported the electrophysiological effects of HCQ or AZM in cardiomyocytes in vitro, our study is the first to evaluate HCQ and AZM in terms of the effect of clinically relevant long-term treatment [22].

In agreement with the reduced cell viability and impaired electrophysiological function, iPSC-CMs also showed altered contractile performance. Treatment with 10 µM AZM or HCQ led to decreased contraction and relaxation time as well as highly varying contraction and relaxation velocities, indicating that treatments with AZM or HCQ at a high concentration over a long time period interfere with the ability of iPSC-CMs to contract in a coordinated manner. In a recent publication, the effects of two cardiotoxic drugs, doxorubicin (DOX) and trastuzumab (TRZ), on the viability and function of iPSC-CMs were reported [41]. Unlike in our study, spontaneous beating frequency and electrical propagation of iPSC-CMs were not affected by DOX and TRZ, but the contraction velocity and displacement (or deformation distance) were reduced. These findings point toward different mechanisms of drug-induced cardiac complications induced by AZM and HCQ compared to DOX and TRZ. The adverse effects induced by DOX and TRZ were proposed to be linked to drug-induced mitochondrial dysfunction and altered cardiac energy metabolism [41]. Based on our results, we assume that the HCQ-induced increase in CV and alteration in contraction may be caused by enhanced expression (or accumulation) of Cx43 and altered gating properties of the sodium channel. Acute treatment with HCQ was reported to have an effect on *I_Na_* with an IC_50_ of 113.9 ± 78.3 µM, which may explain the reduction in the electrical signal transmission observed in the guinea pig heart treated with 10 µM HCQ ex vivo [20]. In our study, we observed no effect of 10 µM HCQ on *I_Na_* after the 7-day treatment, but altered gating properties. This may account for the different effect of HCQ on CV in iPSC-CMs compared to that in the whole heart after acute treatment with 10 µM HCQ. In addition, we cannot exclude the possibilities that these different effects are due to species differences between humans and guinea pigs.

It is worth mentioning that the effects of AZM and HCQ at low concentrations (1 or 3 µM) on cell area, and sarcomere structure of iPSC-CMs were relatively mild but failed to recover to the control level after 7 days of drug washout, suggesting that AZM and HCQ may induce persistent, long-term damage of iPSC-CM structure. In addition, treatment with AZM caused cellular hypertrophy, as shown by increased cell area and higher membrane capacitance (Figure 2B and Appendix A) whereas HCQ (1 or 3 µM) reduced cell area in a concentration-dependent pattern.

Overall, investigation of the individual effects of AZM and HCQ on iPSC-CMs revealed remarkable differences in their influence on the beating frequency, contractile properties, as well as FPDc.

### 3.2. Synergic Effects between AZM and HCQ

Higher mortality rates, significantly increased risks for cardiac arrest [42], and greater QTc prolongation [9,10] were observed in patients treated with HCQ and AZM in combination compared to treatment with either HCQ or AZM. By treating iPSC-CMs with a combination of AZM and HCQ, we confirmed this synergistic effect, which caused a strong reduction in cell viability, sarcomere disorganization, conduction abnormalities, and contractile dysfunction.

On a functional level, changes in contractile activity and electrophysiological properties induced by HCQ and AZM were more pronounced in iPSC-CMs with the combined treatment. In the presence of 10 µM AZM, contraction and relaxation time was already shortened in combination with 1 µM HCQ and were even more reduced with 3 µM and 10 µM HCQ. In addition, the HCQ-induced prolongation of FPDc was further exacerbated in the presence of 10 µM AZM, correlating with greater QTc prolongation in patients receiving HCQ and AZM in combination [9,10]. Similar to FPDc, co-treatment with 10 µM AZM was shown to potentiate the HCQ-induced increase in CV. Of note, documentations of the iPSC-CMs during the 7-day washout period revealed reversibility of the changes in FPDc and CV in some cultures. However, more iPSC-CM cultures with AZM and HCQ combination treatment showed beating arrest compared to treatment with HCQ and AZM alone. These results demonstrate that application of AZM together with HCQ worsens the adverse effects of HCQ to induce contractile and electrophysiological dysfunction in iPSC-CMs. These findings are in line with the increased expression of Nav1.5 and Cx43 in iPSC-CMs, which are induced by HCQ and AZM in a synergistic manner.

In terms of cell viability and structural organization, treatment with 10 µM AZM had no effect on cell viability, but the combination of 10 µM HCQ with 1 µM or 10 µM AZM significantly enhanced the cytotoxicity of 10 µM HCQ, as indicated by lower MTT reduction to formazan and increased LDH activity. This potential of AZM to enhance cytotoxic effects of different drugs was previously demonstrated in cancer cell lines for the combination of AZM with Lansoprazol [43] or gefitinib [44]. Similar to the viability studies, the combination of AZM with HCQ led to the most pronounced reductions in cell area, sarcomere length, and degree of sarcomeric organization, compared to the effects of AZM or HCQ alone.

### 3.3. Mechanistic Evidence of AZM and HCQ Combination

HCQ and AZM are lysosomotropic compounds known to accumulate in lysosomes and to increase lysosomal pH, which is critical for the inhibition of viral infection [45]. Determination of drug levels in iPSC-CMs after the 7-day treatment with AZM or HCQ alone revealed that the cellular levels correlated with the drug concentrations used. Interestingly, cellular levels of AZM were higher when HCQ was present and vice versa, indicating that the combined treatment favors the accumulation of both compounds in iPSC-CMs. Previous study suggested that the ATP-dependent translocase ABCB1 plays an important role in the synergistic effects of AZM and HCQ. ABCB1 is located in the cell membrane and lysosomal membrane, and acts as an AZM-transporter and is known to be inhibited by HCQ [46]. However, involvement of ABCB1 in the synergistic effect of AZM and HCQ in iPSC-CMs is unlikely, as RNA-sequencing data from our group as well others reveal that ABCB1 is not expressed in iPSC-CMs [47,48]. So far, the mechanism for the increased cellular accumulation of AZM and HCQ with combined treatment is unclear.

Activation of integrated stress response pathway and inhibition of autophagosome formation by AZM and HCQ likely explain the strong intracellular accumulation of Nav1.5 and Cx43. Previous studies showed that application of CQ increased the abundance of Cx43 in neonatal rat ventricular myocytes through its lysosomal inhibiting ability and prolongation of Cx43 turnover [49,50]. Remarkably, our study shows that the synergistic effect of AZM and HCQ increased Cx43 expression by 4-fold, which was significantly higher than the increase in Cx43 protein expression observed by treatment with AZM alone. Additionally, 7-day treatment with AZM and HCQ increased protein expression of Nav1.5 but did not increase sodium current density, suggesting that the availability of functional sodium channels on the membrane was not altered despite the intracellular accumulation [22]. As cardiac conduction is determined not only by sodium channel availability but also by gap junction expression and function, our data suggest that the significantly increased expression of the gap junctional protein Cx43 may contribute to the increased CV in iPSC-CMs after the 7-day treatment with HCQ or HCQ and AZM in combination.

Taken together, our results reveal that the more severe effects of the combined treatment with AZM and HCQ on viability, structure, and functionality of iPSC-CMs may be caused by an increased intracellular accumulation of the drugs. The synergistic upregulation of Cx43 protein levels by AZM and HCQ provides first mechanistic evidence for the increased cardiac complications observed with the combination treatment.

### 3.4. Study Limitations

Aiming to gain mechanistic insights for the increased rates of cardiac complications observed for the combined treatment with AZM and HCQ, we characterized the consequences of the two drugs as well as their combination on the viability, structure, and functionality of iPSC-CMs. Despite human iPSC-CMs represent an important in vitro model system to study drug effects on the human heart, different aspects, including the immaturity of the cells and the lack of the multicellular environment, may influence data interpretation, particularly the susceptibility of the cells to drug concentrations. Nevertheless, we believe that human iPSC-CMs are a valuable system for predicting cardiac toxicity of the drug, relevant to human cardiac electrophysiology and dysregulation of a variety of cellular processes. In this study, we applied human iPSC-CMs from four healthy donors without known cardiovascular diseases. Modeling the situation of existing cardiac dysfunctions in patients may require the use of patient-specific iPSC-CMs to extend our study.

## 4. Materials and Methods

### 4.1. Culture and Maintenance of iPSCs

Human iPSC lines used in this study were reprogrammed from somatic cells of four healthy individuals. The cell lines iWTD2.1/2.3 (UMGi001-A clone 1 and clone 3) and iBM76.1/76.3 (UMGi005-A clone 1 and clone 3) were generated from dermal fibroblasts and mesenchymal stem cells, respectively, using STEMCCA lentivirus, and characterized as previously described [47,51]. The cell lines isWT1.13 (UMGi014-C clone 3) and isWT7.22 (UMGi020-B clone 22) were generated from dermal fibroblasts using the integration-free CytoTune-iPS 2.0 Sendai Reprogramming Kit (Thermo Fisher Scientific, Waltham, MA, USA), and characterized previously [52]. The iPSC generation was approved by the Ethics Committee of the University Medical Center Göttingen (approval number: 21/1/11 and 10/9/15) and used following the approval guidelines. To maintain the growth of iPSCs, a chemically defined E8 medium (Thermo Fisher Scientific, Waltham, MA, USA) was used, and cells were cultivated on Geltrex (Thermo Fisher Scientific, Waltham, MA, USA) coated plates at 37 °C with 5% CO_2_. The E8 medium was changed on a daily basis and cells at ~85% confluency were passaged using Versene (Thermo Fisher Scientific, Waltham, MA, USA).

### 4.2. Differentiation of iPSCs into Cardiomyocytes and Drug Treatment

Directed differentiation of iPSCs into cardiomyocytes was induced by modulating the WNT signaling cascade as described [53,54]. In brief, when iPSCs grown on 12-well plates reached 80~90% confluency, the medium was changed from the E8 medium to cardio differentiation medium, which is composed of RPMI 1640 with Glutamax and HEPES (Thermo Fisher Scientific, Waltham, MA, USA), 0.5 mg/mL human recombinant albumin (Sigma-Aldrich, St. Louis, MO, USA), and 0.2 mg/mL L-ascorbic acid 2-phosphate (Sigma-Aldrich, St. Louis, MO, USA). To initiate differentiation, cells were incubated with 4 µM CHIR99021 (a GSK3β inhibitor, Merck Millipore, Darmstadt, Germany) for 48 h followed by incubation with 5 µM IWP2 (a WNT signaling inhibitor, Merck Millipore, Darmstadt, Germany) for an additional 48 h. Thereafter, cells were kept in cardio differentiation medium for four days with medium change every second day. The first beating cells were detected on day 8 post differentiation. From day 8, cells were cultivated in RPMI/B27 medium containing RPMI 1640 with Glutamax and HEPES, supplemented with 2% B27 (Thermo Fisher Scientific, Waltham, MA, USA).

To maintain a long-term culture, iPSC-CMs were replated from 12-well plates into 6-well plates at day 20 post differentiation. Briefly, cells were incubated with 1 mg/mL collagenase B (Worthington Biochemical, Lakewood, USA) for 1 h at 37 °C. Detached iPSC-CM clusters were gently collected into a 15-mL Falcon tube and dissociated with 0.25% trypsin/EDTA (Thermo Fisher Scientific, Waltham, MA, USA) for 8 min at 37 °C. Dissociated iPSC-CMs were resuspended in cardio digestion medium (80% RPMI/B27 medium, 20% fetal calf serum, and 2 µM thiazovivin (Merck Millipore, Darmstadt, Germany)) and cultured in Geltrex-coated 6-well plates at a density of 800,000 cells per well for 24 h. Afterward, iPSC-CMs were cultivated in RPMI/B27 medium.

To perform functional analyses, 70-day-old iPSC-CMs were dissociated again with collagenase B and trypsin stepwise, and replated for different assays. One week after replating, the cells were treated with HCQ and AZM alone or in combination at different concentrations for 7 days, with daily medium change, followed by a 7-day washout period with RPMI/B27 medium (Appendix A). HCQ (EMD Merck Millipore, Darmstadt, Germany) was dissolved in ddH_2_O and AZM (Sigma-Aldrich, St. Louis, MO, USA) was dissolved in DMSO to prepare 10 mM stock solutions, which were aliquoted and stored at −20 °C.

### 4.3. Video-Based Contraction Analysis

Video-based analyses were used to examine drug effects on the contractile parameters of iPSC-CMs. To this end, iPSC-CMs were replated into Geltrex-coated 48-well plates at a density of 60,000 cells per well one week before drug treatment. Videos were obtained using an ORCA Flash 4.0 V3 CMOS camera (Hamamatsu Photonics, Hamamatsu, Japan, 60 FPS, 1024 × 1024 pixels resolution) on days 0 (right before treatment), 1, 3, 5, and 7 of the treatment. Video data were analyzed using the cellular motion analysis software *“Maia”* (QuoData–Quality & Statistics GmbH) to evaluate the beating properties [55]. Analysis settings were: block size 20.3 µm (16 pixels), frameshift 100 ms, and maximum distance shift 8.9 µm (7 pixels). For every condition, videos were obtained from three different wells with two videos on different areas of each well. For analysis, data were normalized to control without drugs of the respective day.

### 4.4. Immunofluorescence Staining

For immunostainings, iPSC-CMs were seeded into Geltrex-coated 12-well or 6-well plates prepared with coverslips at a density of 15,000 or 200,000 cells per well, respectively. After seeding, cells were cultured for 7 days in RPMI/B27 medium before drug treatment. On day 7 (after drug treatment for 7 days) or day 14 (after drug washout for 7 days), cells were washed 2 times for 5 min in relaxation buffer (PBS supplemented with 5 mM EGTA and 5 mM MgCl_2_), followed by 2 times wash with PBS and fixation in ice-cold methanol-acetone (7:3, *v*/*v*) solution for 20 min at −20 °C. Fixed cells were washed 3 times for 5 min with PBS, followed by blocking in 1% BSA (bovine serum albumin) for at least 2 h at 4 °C. For staining, cells were incubated with the following primary antibodies: anti-α-actinin (clone EA-53; 1:500; mouse monoclonal, IgG1, Sigma-Aldrich, St. Louis, MO, USA, 7811), anti-Nav1.5 (1:200; rabbit polyclonal, Alomone Labs, Jerusalem, Israel, ASC-005), and anti-Cx43 (clone 2; 1:1000; mouse monoclonal, IgG1, BD Biosciences, Franklin Lakes, USA, 610061) at 4 °C overnight. Afterward, cells were washed three times with PBS and incubated with the corresponding secondary antibodies (1:1000; anti-rabbit Alexa Fluor 488, Invitrogen, Waltham, MA, USA, A11008; anti-mouse Alexa Fluor 488, Invitrogen, Waltham, MA, USA, A11001; or anti-mouse Alexa Fluor 546, Invitrogen, Waltham, MA, USA, A11030) for 1 h at room temperature. Cell nuclei were counterstained with Hoechst33342 (1:1000; Thermo Fisher Scientific, Waltham, MA, USA) in PBS for 20 min. Coverslips were mounted on glass slides using Fluoromount-G mounting medium (Thermo Fisher Scientific, Waltham, MA, USA). Stained iPSC-CMs were imaged using a fluorescence microscope (Keyence BZ-X700E, Keyence, Osaka, Japan). Quantification of cell area was performed in iPSC-CMs stained for α-actinin using Cell Profiler [56] and manual analysis with FIJI [57]. Sarcomere length was determined manually using FIJI as described previously [36]. The amount of structurally organized iPSC-CMs with evenly distributed intact sarcomeres across the cell body (occupying > 80% of the cell area) and disorganized cells was determined using manual counting.

### 4.5. Multi-Electrode Array

For FP measurement, iPSC-CMs were seeded in the cavity containing electrodes of the Geltrex-coated CytoView 6-well MEA plates (Axion BioSystems, Atlanta, GA, USA). Around 300,000 iPSC-CMs were resuspended in 20 µL cardio digestion medium and seeded in the electrode-containing cavity of the MEA plates. One hour later, an additional 1 mL of medium was added into each well. iPSC-CMs were kept in RPMI/B27 medium for one week before drug treatment. For every batch of experiment, at least two wells of iPSC-CMs from different plates were treated with the same condition to avoid plate variability. Spontaneous FP recordings were carried out using the Maestro Edge equipped with AxIS Navigator software (Axion BioSystems, Atlanta, GA, USA) with a sample rate of 12,500 Hz at 37 °C with 5% CO_2_. From day 0 (right before treatment) to day 14 (last day for washout), FPs were recorded daily for all conditions used (Appendix A). Several key parameters including conduction velocity (CV), corrected FPD_C_ (corrected by Fridericia’s formula), and inter-beat interval were determined using AxIS Navigator, and further analyzed with AxIS Metric Plotting Tool (Axion BioSystems, Atlanta, GA, USA). Spontaneous beating frequency was defined as the reciprocal of averaged inter-beat interval. The mainstream CV values were averaged for one culture.

### 4.6. Automated Patch-Clamp

To investigate the effect of high concentrations of HCQ and AZM on the function of sodium channel, the properties of *I_Na_* were examined in iPSC-CMs treated with 10 µM HCQ alone, 10 µM AZM alone, or their combination, respectively. The drug treatment lasted for 7 days with daily medium change, and iPSC-CMs kept in RPMI/B27 medium served as control. Recording of *I_Na_* was performed using the Patchliner Quattro (Nanion Technologies GmbH, Munich, Germany) with low resistance NPC-16 chips at room temperature as described previously [53,58,59]. In brief, iPSC-CMs were dissociated gently into single cells. Capture of single cells and formation of whole-cell configuration were processed automatically by Patchliner. From a holding potential of −100 mV, *I_Na_* was recorded under pulses ranging from −90 to +70 mV for 20 ms in 5 mV increment with an interval of 2 s. Currents were sampled at 25 kHz and low-pass-filtered at 2.9 kHz.

### 4.7. Western Blot

Three-month-old iPSC-CMs were treated with 10 µM HCQ, or 10 µM AZM, or the combination of HCQ and AZM for seven days, snap-frozen in liquid nitrogen and stored at −80 °C. To detect the expression of specific proteins, cells were lysed by homogenization in RIPA buffer (150 mM NaCl, 50 mM Tris, 1.0% NP-40, 0.5% sodium deoxycholate, 0.1% SDS, 1 mM EDTA, 10 mM NaF, and 1 mM PMSF), supplemented with protease (cOmplete mini, EDTA-free, Roche, Basel, Switzerland) and phosphatase (PhosSTOP, Roche, Basel, Switzerland) inhibitors and incubated for 30 min at 4 °C with gentle rotation. Cell homogenates were clarified by centrifugation at 14,000 rpm for 20 min at 4 °C and protein concentration was measured using a BCA assay following the manufacturer’s instruction. Proteins (30 µg/sample) were subjected to SDS-PAGE using a 4–15% gradient gel (BioRad, Hercules, CA, USA) and transferred onto nitrocellulose membranes. Membranes were blocked in 5% milk in TBS-T for 30–45 min at room temperature and probed with anti-Cx43 (clone 4E6.2; 1:1000; mouse monoclonal, Merck Millipore, Darmstadt, Germany, MAB3067), anti-Nav1.5 (1:200; rabbit polyclonal, Alomone Labs, Jerusalem, Israel, ASC-005), or anti-EEF2 (1:5000; rabbit polyclonal, Abcam, Cambridge, UK, ab40812) antibodies at 4 °C overnight, followed by incubation with horseradish peroxidase-conjugated secondary antibodies goat anti-mouse (1:10,000; Sigma Aldrich, St. Louis, MO, USA, A2304) or goat anti-rabbit (1:10,000; Cell Signaling, Danvers, MA, USA, 7074S) for 1 h at room temperature. Proteins were visualized by chemiluminescence using the Super Signal West Dura Chemiluminescent Substrate kit in combination with the Fusion FX Spectra Imaging System (Peqlab, VWR, Radnor, PA, USA). Densitometry analyses of the immunoblots were performed using ImageJ software and the intensity of individual bands was normalized to EEF2.

### 4.8. Lactate Dehydrogenase Measurement

Measurement of LDH activity was performed using LDH assay kit (Abcam, Cambridge, UK, ab102526) according to the manufacturer’s instructions in supernatants of iPSC-CM cultures after 7 days of drug treatment and after subsequent 7 days of drug washout. Briefly, 50 µL of cell supernatant was mixed with 50 µL substrate solution in a 96 well plate. Absorption was measured at 450 nm in a kinetic mode, every 2 min for 60 min (Biotek Synergy HTX, Biotek Instruments—Agilent, Santa Clara, CA, USA). LDH activity was calculated based on a standard curve according to the manufacturer’s instructions (Equation (1)).

Equation (1): Calculation of LDH activity
(1)$$LDH\;activity\;\left[\frac{\mathrm{mU}}{\mathrm{mL}}\right]=\left(\frac{Amount\;of\;NADH\;in\;sample\;calc.\;from\;standard\;curve\;\left[\mathrm{nmol}\right]}{reaction\;time\;[\min]\;\times \;Sample\;volume\;\left[\mathrm{mL}\right]}\right)\times \;Dilution\;factor$$

### 4.9. MTT Assay

Cell viability was determined using MTT assay kit (Merck Millipore, Darmstadt, Germany, CT02) according to the manufacturer’s instructions. After drug treatment as well as after drug washout, cells were washed twice with pre-warmed PBS and incubated in 200 µL RPMI/B27 medium per well with 0.5 mg/mL MTT for 2 h at 37 °C. Subsequently, 300 µL of isopropanol with 0.04 N HCl was added and samples were mixed thoroughly by pipetting to facilitate cell lysis and the dissolving of formazan. Absorbance was measured at 570 nm (formazan) and 630 nm (reference) using plate reader (Biotek Synergy HTX, Biotek Instruments—Agilent, Santa Clara, CA, USA). Viability was calculated as A_570_–A_630_.

### 4.10. Determination of HCQ and AZM Concentrations in Cell Lysates

Intracellular drug accumulation was determined from cell lysates of the MTT assay using mass spectrometry. After MTT measurement, cell lysates were stored at −20 °C for 1–4 days prior to detection. The stability of HCQ and AZM under these conditions was confirmed for up to 7 days at −20 °C. Total of 25 µL of fresh or thawed cell lysates were diluted with 225 µL of 2 mM ammonium acetate buffer, vortexed and centrifuged for 10 min (14,000 rpm). About 10 µL of the clear supernatants were injected into the LC-MS/MS, which consists of an UltiMate3000 pump, an autosampler (Dionex, ThermoScientific, St. Louis, MO, USA) and an API 4000 Tandem mass spectrometer (ABSciex, Darmstadt, Germany) using positive electrospray ionization (ESI+; 4500 V). HCQ and AZM were determined by a Synergi 4µ HydroRP 80A column 150 mm × 3.0 mm (Phenomenex, Torrance, CA, USA) using a binary gradient with 2 mM ammonium acetate buffer and acetonitrile and a flow rate of 0.5 mL/min. The resulting retention times were 3.0 min for HCQ and 3.2 min for AZM. HCQ and AZM were measured using the multiple reaction monitoring mode with nitrogen as collision gas. The method was suitable for the quantification of HCQ and AZM in cell lysates over the range from 20 to 1000 ng/mL. Samples with higher concentrations were diluted.

### 4.11. Statistics

Results about cell area are presented as median ± 95% CI and results for the other parameters are presented as mean ± standard error of the mean (SEM). Statistical analysis was performed with GraphPad Prism 9. One-way ANOVA with Tukey’s multiple comparison was used for cell viability, cell area, sarcomere length, contractility property, protein expression level, and drug accumulation data. Two-way ANOVA with Bonferroni post-hoc test was used for MEA assay-based FPDc, CV and beating rate data, as well as Patchliner assay-based *I_Na_* data. *p* < 0.05 was considered statistically significant.

## 5. Conclusions

Through the systematic investigation of the effects of AZM and HCQ individually as well as in combination, we show that these two drugs have adverse effects on the viability, structure, and functionality of human cardiomyocytes using the in vitro iPSC-CM system. These adverse effects get more severe when AZM and HCQ are applied in combination, thus recapitulating the higher rates of cardiac complications observed with the AZM/HCQ combination treatment in clinical use. The synergistic adverse effects of AZM and HCQ in iPSC-CMs are likely driven by the increased intracellular accumulation of the drugs when applied in combination. Furthermore, we provide evidence that the HCQ-induced increase in conduction velocity might be caused by elevated levels of Cx43, which further increase in combination with AZM.

## Figures and Tables

**Figure 1 pharmaceuticals-15-00220-f001:**
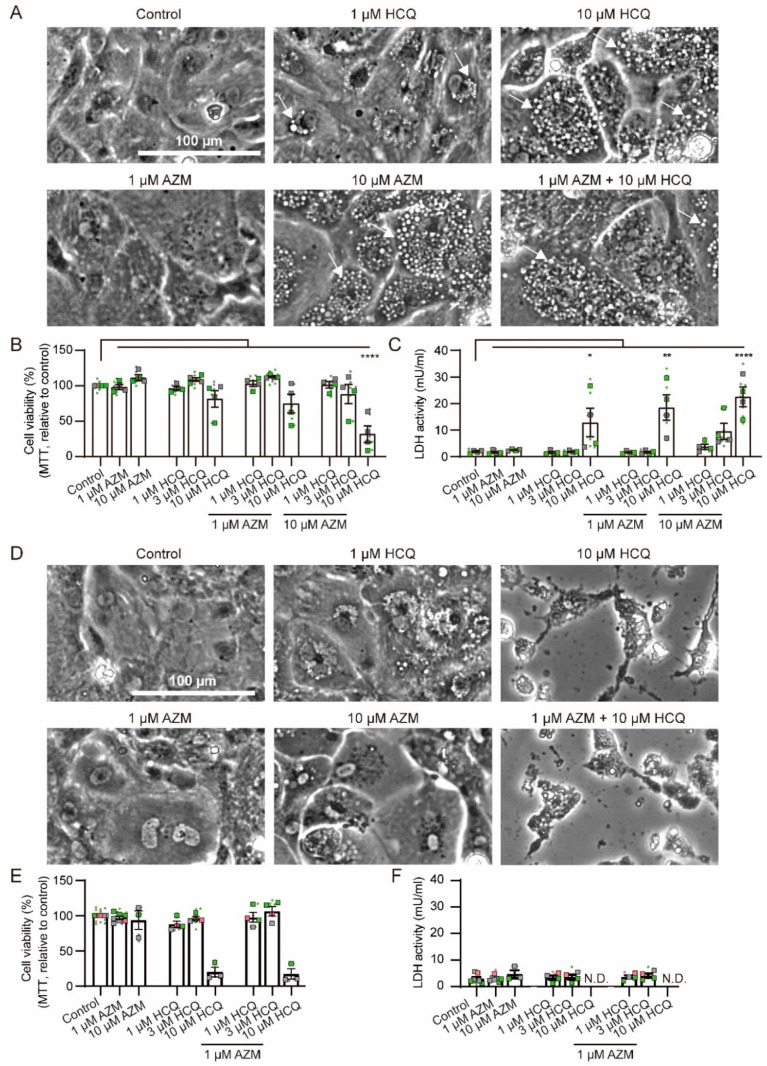
Morphological changes and cytotoxicity in iPSC-CMs treated with HCQ and AZM. (**A**) Representative brightfield images depicting morphology of iPSC-CMs after 7-day treatments with HCQ and AZM in different concentrations. Arrows indicate formation of vacuoles. Scale bar: 100 µm. (**B**) Cell viability after 7-day drug treatment as determined by measurement of formazan formation in the MTT assay. (**C**) LDH activity detected in cell supernatants after 7-day drug treatment. (**D**) Representative brightfield images depicting morphology of iPSC-CMs after 7-day drug treatment and 7-day washout period. Even after washout, iPSC-CMs treated with a combination of high concentrations of AZM and HCQ show severe morphological changes and increased cell death. Scale bar: 100 µm. (**E**) Cell viability after 7-day drug washout as determined by using the MTT assay. (**F**) LDH activity detected in supernatants after 7-day drug washout. Data represent technical replicates (points) and means (squares) of each experiment, *N* = 3–7 independent experiments using iPSC-CMs from 3 healthy donors (iBM76.1, iBM76.3 in green; iWTD2.1, iWTD2.3 in grey, isWT7.22 in pink). Lines and errors show overall mean and SEM. Statistical analysis was performed using one-way ANOVA and Tukey’s multiple comparison test. * *p* < 0.05, ** *p* < 0.01, **** *p* < 0.0001. N.D.—not determined.

**Figure 2 pharmaceuticals-15-00220-f002:**
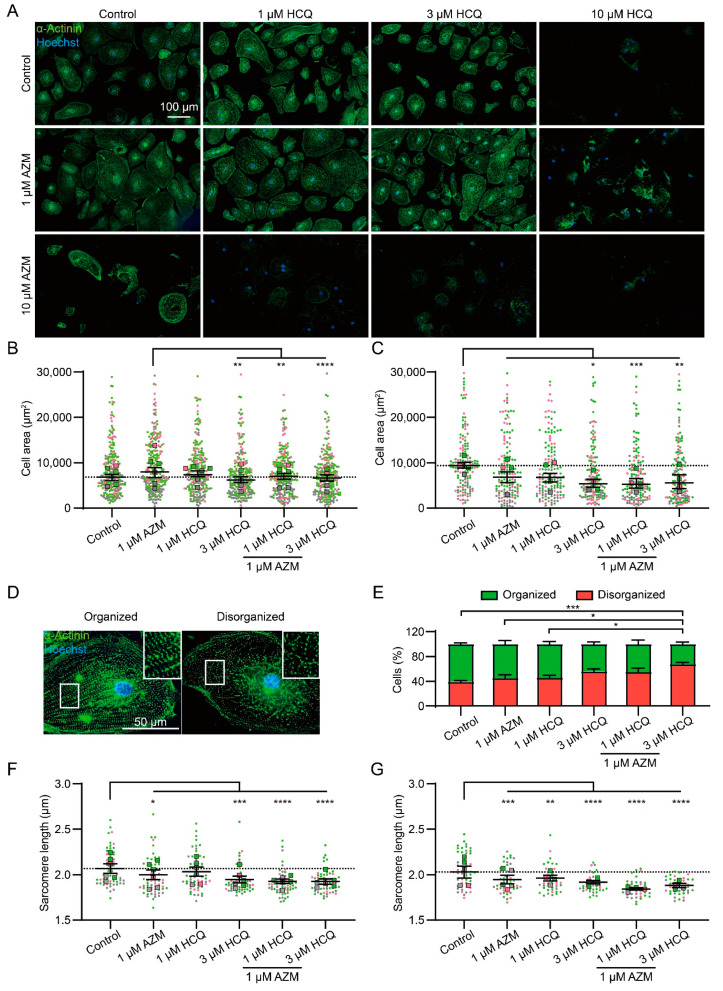
HCQ and AZM cause sarcomeric disorganization in iPSC-CMs. (**A**) Representative images of α-actinin immunostained iPSC-CMs treated with different concentrations of HCQ and AZM for 7 days. (**B**,**C**) Analysis of cell areas after 7-day drug treatment (**B**) and after subsequent 7-day washout (**C**). A total of *n* = 160–240 cells (40 per experiment) from 6 (**B**) or 4 (**C**) independent experiments per condition were analyzed. Points represent values of single cells and squares the median values of individual experiments. Lines indicate median and 95% CI of the overall population. (**D**) Representative images of structurally organized and disorganized iPSC-CMs after drug treatment for 7 days. (**E**) Percentage of structurally organized and disorganized iPSC-CMs after 7-day drug treatment. Mean and SEM of 5 independent experiments (*n* = 96–272 cells analyzed per condition from each experiment, same number of cells at different conditions analyzed within one experiment) are shown. (**F**,**G**) Sarcomere length after 7-day drug treatment (**F**) and after 7-day washout (**G**). Mean and SEM of *n* = 50–60 cells (10 per experiment) from 6 (**F**) or 5 (**G**) independent experiments are shown. Data plots in (**F**), and (**G**) show technical replicates (dots) and mean values (squares). Colors indicate iPSC-CM differentiations from 3 healthy donors (iBM76.1, iBM76.3 in green; iWTD2.1, iWTD2.3 in grey, isWT7.22 in pink). Statistical analysis was performed using one-way ANOVA and Tukey’s multiple comparison test. * *p* < 0.05, ** *p* < 0.01, *** *p* < 0.001, **** *p* < 0.0001.

**Figure 3 pharmaceuticals-15-00220-f003:**
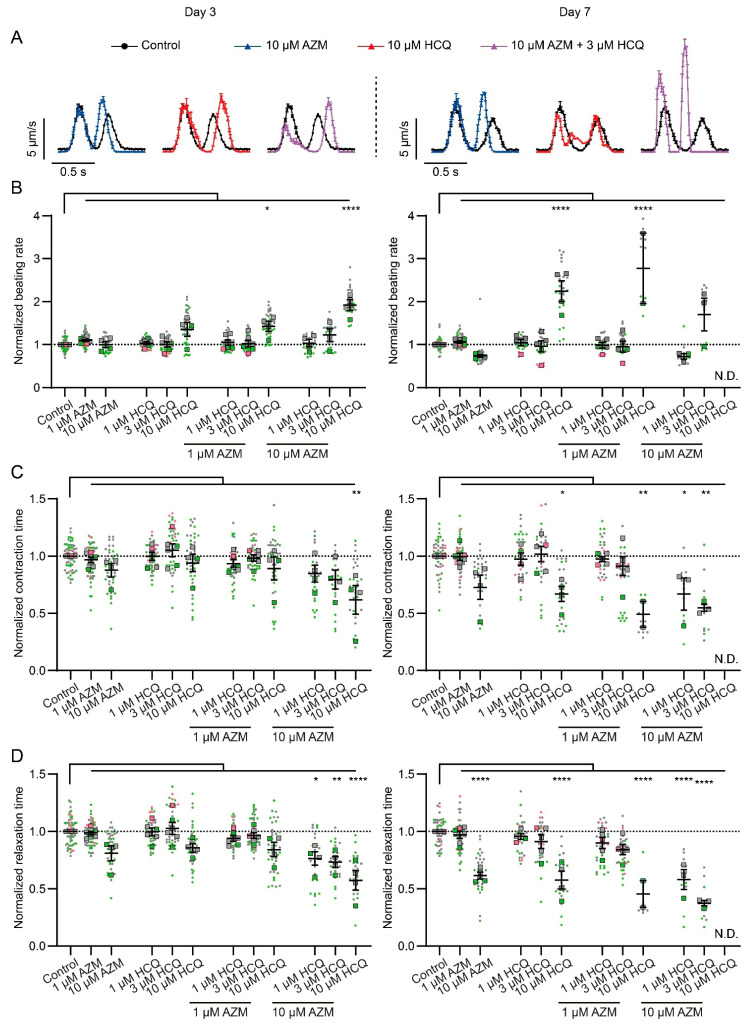
Contractile dysfunctions in iPSC-CMs treated with HCQ and AZM. (**A**) Representative motion traces observed in iPSC-CMs using vector-based quantification on treatment day 3 (**left**) and 7 (**right**). Values represent mean and SEM of motions from aligned contraction-relaxation cycles of a representative video. (**B**–**D**) Effects of AZM and HCQ alone as well as their combination on the beating rate (**B**), contraction time (**C**), and relaxation time (**D**) on treatment day 3 (**left**) and 7 (**right**). Data represent technical replicates (points, *n* = 9–54 videos per condition) and means (squares) of each experiment, *N* = 4–6 independent experiments using iPSC-CMs from 3 healthy donors (iBM76.1, iBM76.3 in green; iWTD2.1, iWTD2.3 in grey, isWT7.22 in pink). Due to the toxic effects of HCQ or AZM at higher concentrations or in combination, fewer videos could be analyzed under these conditions. Lines show overall mean values and SEM. Statistical analysis based on the mean values of the individual experiments using one-way ANOVA and Tukey’s multiple comparison test. * *p* < 0.05, ** *p* < 0.01, **** *p* < 0.0001.

**Figure 4 pharmaceuticals-15-00220-f004:**
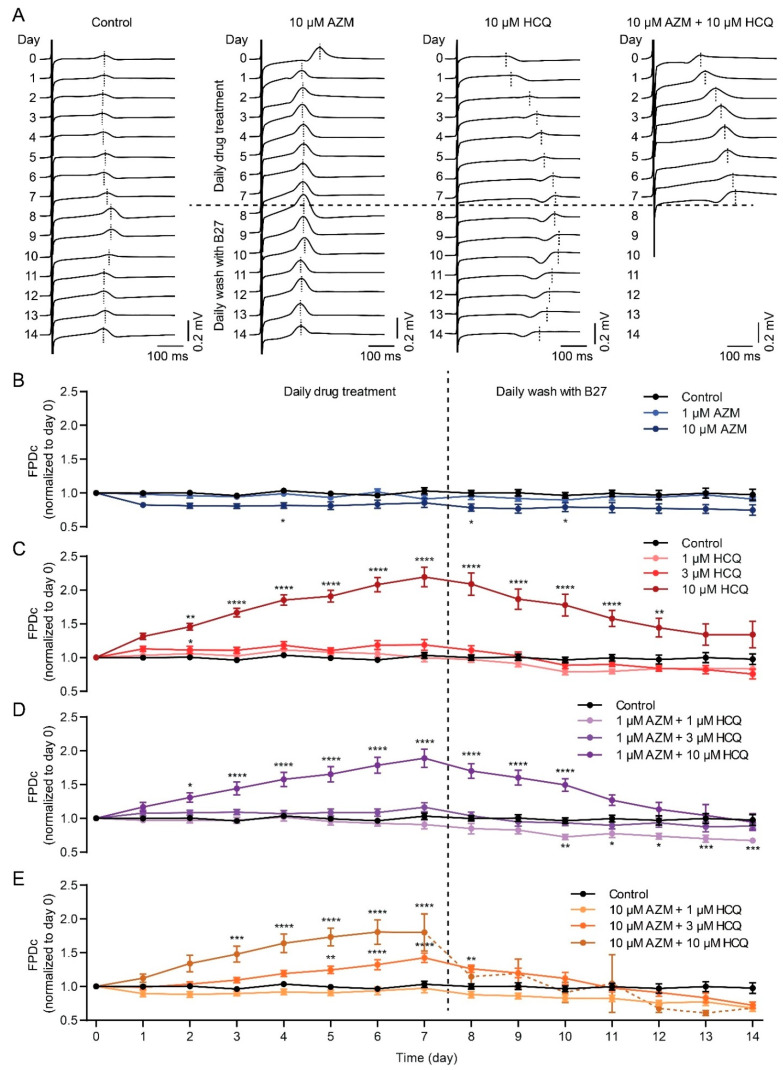
Effects of HCQ and AZM on the field potential duration of iPSC-CMs. (**A**) Representative recordings of extracellular FP in spontaneous beating iPSC-CMs under different treatment conditions. iPSC-CMs treated with 10 µM AZM and 10 µM HCQ in combination stopped beating at day 8 (one day after initiation of washout). (**B**,**C**) Effect of AZM (**B**) or HCQ (**C**) on the corrected FPD (FPDc, normalized to day 0) during 7-day treatment and subsequent 7-day washout. (**D**,**E**) Effects of HCQ (1, 3, and 10 µM) combined with 1 µM AZM (**D**) or 10 µM AZM (**E**) on FPDc during 7-day treatment and following 7-day washout. iPSC-CMs derived from four donors were used for MEA recording. For the initial recording (day 0), 10 ≤ *n* ≤ 13 for all conditions. Spontaneous beating states of iPSC-CMs are listed in Appendix A. Two-way ANOVA with Bonferroni post-hoc test was used for statistical evaluation (* *p* < 0.05, ** *p* < 0.01, *** *p* < 0.001, and **** *p* < 0.0001).

**Figure 5 pharmaceuticals-15-00220-f005:**
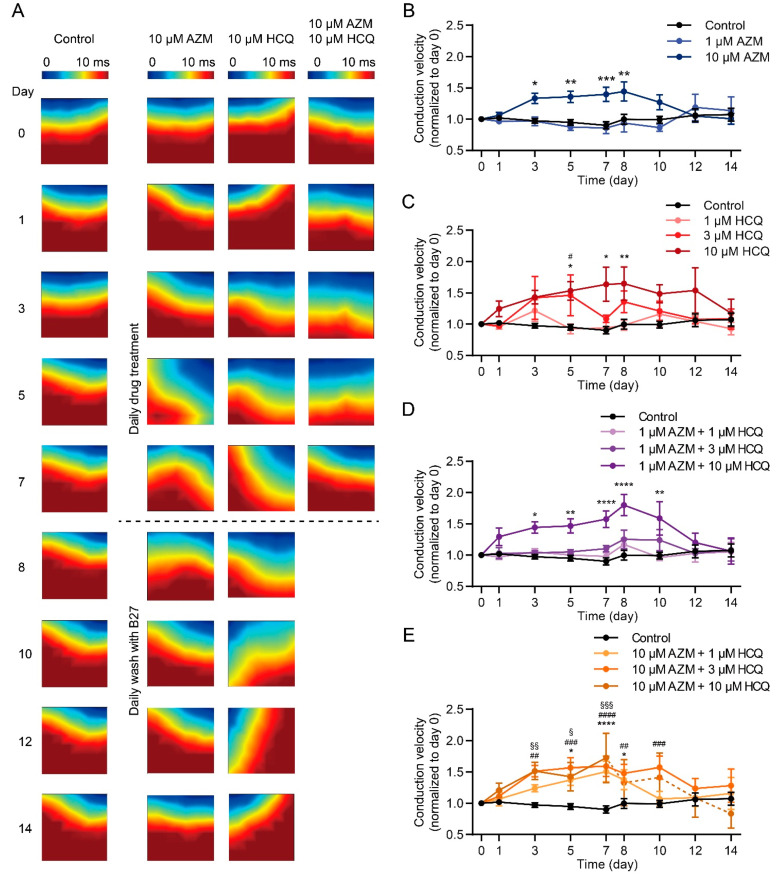
Changes in conduction trajectory and augmented CV in iPSC-CMs treated with AZM and HCQ alone and in combination. (**A**) Representative heatmaps illustrating the conduction trajectories of electrical signals in iPSC-CMs under different conditions during 7-day drug treatment and following 7-day washout. Due to cell death, no signal was captured in cells treated with 10 µM HCQ and 10 µM AZM in combination in the washout period. (**B**,**C**) CV of iPSC-CMs treated with AZM (**B**, *: 10 µM AZM vs. control) and HCQ (**C**, ^#^: 3 µM HCQ vs. control, *: 10 µM HCQ vs. control) for 7 days and following washout for 7 days (normalized to day 0). (**D**,**E**) CV of iPSC-CMs treated with 1 µM (**D**, *: 1 µM AZM + 10 µM HCQ vs. control) and 10 µM AZM (**E**, *: 10 µM AZM + 1 µM HCQ vs. control, ^#^: 10 µM AZM + 3 µM HCQ vs. control, ^§^: 10 µM AZM + 10 µM HCQ vs. control) combined with HCQ (1, 3, and 10 µM) during 7-day treatment and following 7-day washout. iPSC-CMs derived from four donors were used for MEA recording. Spontaneous beating states of iPSC-CMs are listed in Appendix A. Two-way ANOVA with Bonferroni post-hoc test was used (*^,§,#^ *p* < 0.05, **^,§§,##^ *p* < 0.01, ***^,§§§,###^ *p* < 0.001, and ****^,####^ *p* < 0.0001).

**Figure 6 pharmaceuticals-15-00220-f006:**
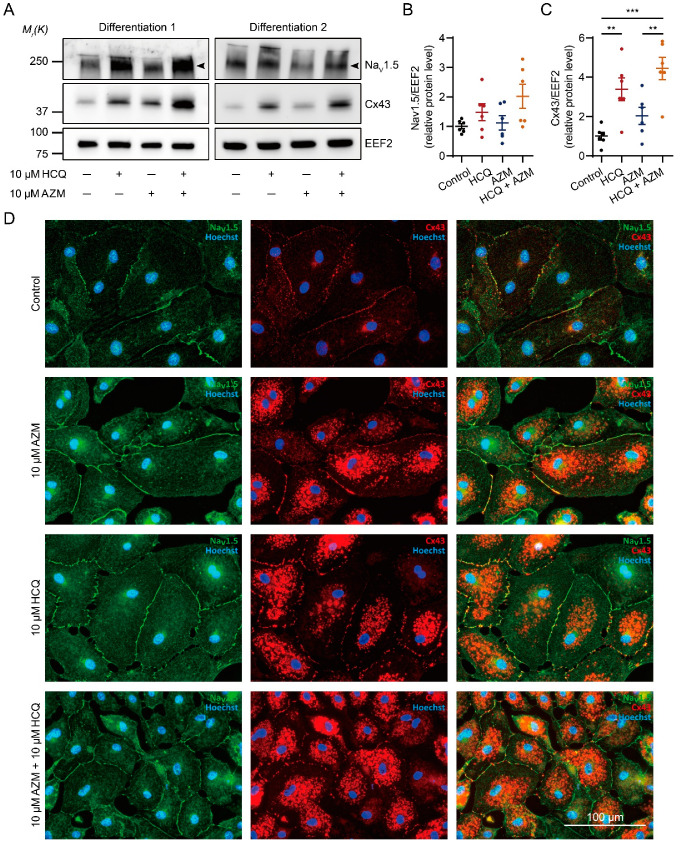
Nav1.5 and Cx43 protein expression in iPSC-CMs treated with HCQ and/or AZM. (**A**) Two representative Western blots showing the expression of Nav1.5 and Cx43 in iPSC-CMs under different drug treatment conditions. (**B**) Quantitation of protein expression levels of Nav1.5 in iPSC-CMs under different conditions; *N* = 6 independent differentiations. (**C**) Quantitation of protein expression of Cx43 in iPSC-CMs under different drug treatment conditions; *N* = 6 independent differentiations. (**D**) Representative images showing immunostaining for Nav1.5 (green) and Cx43 (red) in iPSC-CMs under different drug treatment conditions. Cell nuclei are shown in blue (Hoechst). Statistical evaluation was performed using one-way ANOVA with Tukey’s multiple comparison test (** *p* < 0.01, and *** *p* < 0.001).

**Figure 7 pharmaceuticals-15-00220-f007:**
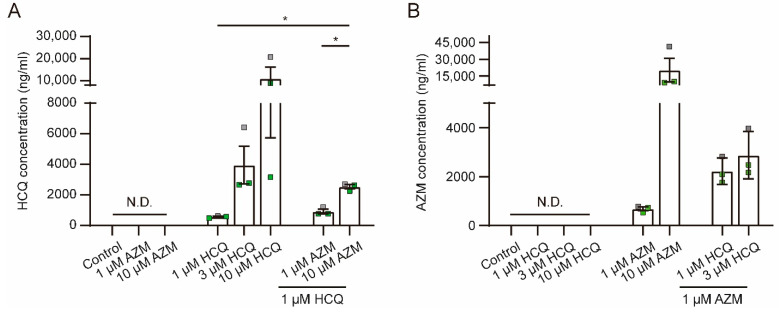
Accumulation of HCQ and AZM in iPSC-CMs after the 7-day treatment. (**A**,**B**) Concentrations of HCQ (**A**) and AZM (**B**) in cell lysates from iPSC-CMs after the 7-day treatment with HCQ and AZM at different conditions, determined using mass spectrometry. Data represent mean and SEM of *N* = 3 independent experiments, performed with iPSC-CMs from 2 healthy donors (iBM76.1, iBM76.3 in green; iWTD2.1 in grey). N.D., below detection limit. Statistical evaluation was performed using one-way ANOVA with Tukey’s multiple comparison test (* *p* < 0.05).

## Data Availability

Data is contained within the article or Appendix A.

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
