# Peer review of "Synergistic Adverse Effects of Azithromycin and Hydroxychloroquine on Human Cardiomyocytes at a Clinically Relevant Treatment Duration"

_pharmaceuticals, 2022, doi:10.3390/ph15020220_

Round 1

Reviewer 1 Report

The study done by Wener eta al to determine the synergistic adverse effects of AZM and HCQ is valid and provide some interesting information.

However, authors need to clarify on certain issues to improve the credibility of the outcome.

  1. Authors need to briefly add about the duration of this combination therapy, why they selected 7 days treatment period and they term it as clinically relevant treatment duration in abstract, how?
  2. The results section starts with reasoning, that is not required, directly they can start from the outcomes instead of developing a background.
  3. Authors need to explain the rationale for the 7 days of washout period.
  4. How concentrations of HCQ and AZM selected?
  5. Figure 1 with morphological changes not clear, need to highlight the changes that has happened due to interventions.
  6. The cell viability was shown to be around 10% with HCQ (10) which is same with AZM (1), there is no difference between them, so how authors say it is synergistic adverse effects? E
  7. Authors should use a scale of 1 unit to better present the result of F figure 1. All values are less than 10, need to expand the figure bars.
  8. The LADH activity in C figure of 1 for control and AZM all doses are similar that means there is no toxicity caused by AZM, while HCQ with or without showed same result, not convinced again for synergistic adverse effects.
  9. Authors need to do the English check of the manuscript to improve the understanding the issues raised in the manuscript.

Author Response

Response to Reviewer 1:

The study done by Wener eta al to determine the synergistic adverse effects of AZM and HCQ is valid and provide some interesting information.

However, authors need to clarify on certain issues to improve the credibility of the outcome.

  1. Authors need to briefly add about the duration of this combination therapy, why they selected 7 days treatment period and they term it as clinically relevant treatment duration in abstract, how?

We thank the reviewer for mentioning this important issue. We moved a short section from the discussion part to the introduction and added further details to better explain the rationale for drug concentrations and the treatment duration in our study and the use of the drugs in the clinic. Please see lines 85-105 in the revised manuscript.

2. The results section starts with reasoning, that is not required, directly they can start from the outcomes instead of developing a background.

We thank the reviewer for this suggestion. In the revised manuscript, we removed this section from the beginning of the Results section and transferred to the end of the Introduction part (addressed in point 1).

3. Authors need to explain the rationale for the 7 days of washout period.

We are grateful to the reviewer for this excellent suggestion. To address this point, we added the following sentence to the Introduction part: “The duration of the washout phase was determined based on clinical case reports describing the reversibility of HCQ-induced QT prolongation 4-7 days after drug withdrawal [24,25].” Please see lines 85-105 in the revised manuscript.

  1. O'Laughlin, J.P.; Mehta, P.H.; Wong, B.C. Life Threatening Severe QTc Prolongation in patient with systemic lupus erythematosus due to hydroxychloroquine. Case Rep Cardiol 2016, 4626279, doi:10.1155/2016/4626279.
  2. Chen, C.Y.; Wang, F.L.; Lin, C.C. Chronic hydroxychloroquine use associated with QT prolongation and refractory ventricular arrhythmia. Clin Toxicol (Phila) 2006, 44, 173-175, doi:10.1080/15563650500514558.

4. How concentrations of HCQ and AZM selected?

Together with the points 1-3, the rationale for the concentrations of HCQ and AZM used is now described in details in the last paragraph of the Introduction part. Please see lines 85-105 in the revised manuscript.

5. Figure 1 with morphological changes not clear, need to highlight the changes that has happened due to interventions.

We appreciate the suggestion of the reviewer. We replaced the microscopy images in Figure 1 with images showing regions of interest in higher magnification. Furthermore, we highlighted the formation of vacuoles with arrows. The original images of Figure 1A, D were moved to Supplementary Figure 2A, B.

6. The cell viability was shown to be around 10% with HCQ (10) which is same with AZM (1), there is no difference between them, so how authors say it is synergistic adverse effects? E

Thank you for mentioning this point. As described in the results section, we observed a reduction in cell viability (MTT turnover to formazan, Fig. 1B) after treatment with 10 µM HCQ (82 ± 12%), but not with 1 µM or 3 µM HCQ. Importantly, cell viability was lower after treatment with 1 µM AZM in combination with 10 µM HCQ (75 ± 13%), and further decreased by 10 µM AZM in combination with 10 µM HCQ (31 ± 12%), although monotreatment with 1 µM AZM (98 ± 4%) or 10 µM AZM (111 ± 4%) had no effect in viability. Based on these data, we conclude that AZM potentiates the cardiotoxic effect of HCQ. Numbers are given in Mean ± SEM.

7.Authors should use a scale of 1 unit to better present the result of F figure 1. All values are less than 10, need to expand the figure bars.

We thank the reviewer for this comment. We chose the Y-axis scale in Figure 1F in this range to be consistent with the graph shown in Figure 1C. Identical scales in both graphs allow the readers to directly compare the effect of HCQ and AZM after 7-day treatment (Fig. 1C) and subsequent 7-day washout (Fig. 1F).

8. The LADH activity in C figure of 1 for control and AZM all doses are similar that means there is no toxicity caused by AZM, while HCQ with or without showed same result, not convinced again for synergistic adverse effects.

We agree with the reviewer that the LDH activity was not significantly affected 1 µM AZM (1.6 ± 0.3 mU/ml) or 10 µM AZM (2.5 ± 0.2 mU/ml) when compared to the control (1.8 ± 0.3 mU/ml). However, as shown in Figure 1C, treatment with 10 µM HCQ led to increased LDH activity (13 ± 5 mU/ml) in the supernatant, in comparison to control (1.8 ± 0.3 mU/ml). This was further increased when HCQ was combined with 1 µM AZM (18 ± 5 mU/ml) or 10 µM AZM (22 ± 4 mU/ml). Furthermore, LDH activity in the supernatant was increased when 3 µM HCQ were combined with 10 µM AZM (9.6 ± 3 mU/ml), in comparison to 3 µM HCQ alone (1.8 ± 0.3 mU/ml). Therefore, we conclude that AZM potentiates the cardiotoxic activity of HCQ and thus acts in a synergistic manner.

9. Authors need to do the English check of the manuscript to improve the understanding the issues raised in the manuscript.

We thank the reviewer for this suggestion and carefully checked the manuscript again.

Reviewer 2 Report

This manuscript written by Li and co-authors investigated the effects of hydroxychloroquine (HCQ), azithromycin (AZM) and their combination on the structure and functionality of cardiomyocytes (CMs) and the underlying mechanisms. The authors demonstrated synergistic adverse effects of AZM and HCQ, as the combined treatment led to greater cardiotoxicity, among other more serious effects, than observed with a single drug. The manuscript is clearly written, the experimental data are sound, accurate and very well illustrated. The present manuscript can be accepted as is.

Author Response

We thank the reviewer for taking time to review our manuscript.

Reviewer 3 Report

This interesting study aimed at investigating the effects of hydroxychloroquine, azithromycin, and their combination on the structure and functionality of cardiomyocytes, exploring the underlying mechanisms of cardiotoxicity in an in vitro human cardiomyocyte model system. The manuscript is well written, the results are clearly described and sufficiently discussed. I have no relevant comments, I only suggest the authors to provide a list of abbreviations

Author Response

We thank the review for the suggestion. We now include a list of abbreviations in the revised manuscript.